# A Novel Pipeline Corrosion Monitoring Method Based on Piezoelectric Active Sensing and CNN

**DOI:** 10.3390/s23020855

**Published:** 2023-01-11

**Authors:** Dan Yang, Xinyi Zhang, Ti Zhou, Tao Wang, Jiahui Li

**Affiliations:** 1Key Laboratory for Metallurgical Equipment and Control of Ministry of Education, Wuhan University of Science and Technology, Wuhan 430081, China; 2Hubei Key Laboratory of Mechanical Transmission and Manufacturing Engineering, Wuhan University of Science and Technology, Wuhan 430081, China; 3Precision Manufacturing Institute, Wuhan University of Science and Technology, Wuhan 430081, China; 4Wuhan Digital Engineering Institute, Wuhan 430074, China

**Keywords:** pipeline corrosion, time reversal, convolutional neural network, wavelet packet

## Abstract

In this study, a piezoelectric active sensing-based time reversal method was investigated for monitoring pipeline internal corrosion. An effective method that combines wavelet packet energy with a Convolutional Neural Network (CNN) was proposed to identify the internal corrosion status of pipelines. Two lead zirconate titanate (PZT) patches were pasted on the outer surface of the pipeline as actuators and sensors to generate and receive ultrasonic signals propagating through the inner wall of the pipeline. Then, the time reversal technique was employed to reverse the received response signal in the time domain, and then to retransmit it as an excitation signal to obtain the focused signal. Afterward, the wavelet packet transform was used to decompose the focused signal, and the wavelet packet energy (WPE) with large components was extracted as the input of the CNN model to rapidly identify the corrosion degree inside the pipeline. The corrosion experiments were conducted to verify the correctness of the proposed method. The occurrence and development of corrosion in pipelines were generated by electrochemical corrosion, and nine different depths of corrosion were imposed on the sample pipeline. The experimental results indicated that the classification accuracy exceeded 99.01%. Therefore, this method can quantitatively monitor the corrosion status of pipelines and can pinpoint the internal corrosion degree of pipelines promptly and accurately. The WPE-CNN model in combination with the proposed time reversal method has high application potential for monitoring pipeline internal corrosion.

## 1. Introduction

Pipelines are the main choice for long-distance transportation of a large number of petroleum products and natural gas due to their high security, strong transmission capacity, low operating costs, and other advantages [1]. However, pipelines are faced with various threats, among which inner wall corrosion is one of the serious threats [2,3,4]. Rupture, leakage, and other problems caused by pipeline inner wall corrosion will lead to property loss and environmental pollution, and even endanger human life [5]. Therefore, it is of great practical significance and economic value to investigate the internal corrosion of pipelines through continuous and regular observations [6,7].

Traditional non-destructive testing methods (NDT) for pipeline corrosion include visual inspection, magnetic flux leakage detection, eddy current detection, ultrasonic tomography, and X-ray technology [8,9,10,11,12]. Visual inspection relies on the expertise of inspectors, and thus, its reliability of results is not guaranteed. Magnetic flux leakage detection requires magnetization of the pipeline, and far-field eddy current detection depends on the formation of eddy currents in the pipeline, thus, they are only appropriate for detecting ferromagnetic materials and conductive materials. These two detection methods have a limited detection range and are inappropriate for the remote detection of oil and gas transportation pipelines. The radiation method is extremely harmful to human health, and the operation is complex and costly [13]. The aforementioned NDT technologies usually detect pipelines at regular intervals, which makes it difficult to achieve real-time monitoring of pipelines. Instead of traditional inspection techniques, some sophisticated and efficient methods have been proposed as well. For example, Tan et al. proposed a method for monitoring pipeline corrosion using distributed fiber optic sensors, which can efficiently monitor, locate, visualize, and quantify pipeline corrosion online [14]. However, the detection process involves a lot of processing, such as additive averaging of signals, scanning of frequencies, and tracking of phases, and hence, it requires a long measurement time. Colvalkar et al. utilized a pipeline inspection robot to detect and identify various pipeline anomalies on the inner surface of the pipeline using Image Processing (I.P) and Machine Vision (M.V). The pipeline anomalies may include blockages, bores, cracks, and corrosion [15]. The robot provides real-time monitoring of the environment as it moves through the pipeline. Defect detection and identification are carried out through machine vision technology. However, when there are many bends in the pipe, the friction between the cable and the pipe wall becomes substantial, which can seriously affect the maximum travel distance of the robot during operation, and also can pose a series of problems such as reliability. Therefore, it is vital to find a reliable, real-time, simple, widely used, harmless to the human body, and low-cost method for monitoring pipeline internal corrosion.

The active sensing approach using surface-mounted or embedded transducers has displayed great potential for the Structural Health Monitoring (SHM) of mechanical and civil structures in real time [16,17,18]. The principle of this approach is based on measuring the property changes of the propagating wave, occurring due to structural damage, by using a pair of sensor transducers or a deployed sensor network. Lead zirconate titanate (PZT) is a commonly used piezoelectric material with dual drive and sensing capabilities and a wide bandwidth and is usually utilized for stress wave generation [19,20] and detection [21,22]. PZT is cheap, easy to produce, small, and light, which can be permanently arranged on the structure to collect the feedback information of the structure in real time [23,24]. The PZT-enabled active sensing approach provides a basis for real-time health monitoring of pipeline structures [25,26,27]. Du et al. employed a stress-waves-based active sensing method to detect real-time damage location for multi-crack pipes [28]. Zhu et al. utilized the PZT probe for impact detection and for positioning submarine pipelines [29]. Zhang et al. put forward a Gaussian Mixture Model–Hidden Markov Model (GMM-HMM) method to detect pipeline leakage and crack depth by extracting the time-domain damage index and frequency-domain damage index from signals collected by PZT sensors [30]. However, most of the working environments of pipelines are exposed to harsh environments. In this case, the method of real-time monitoring of pipeline structure damage based on PZT active sensing is easily affected by noise.

In recent years, the time reversal technique has been extensively used in many fields due to its self-adaptive focus on the solid and its high signal-to-noise ratio. In the time-reversal approach, the sensor-received responding signal is first reversed in the time domain, and then the reversed signal is sent out as an excitation signal and the focused signal is finally received [31]. Due to its strong anti-noise capability, the time reversal method has been broadly used in the health monitoring of pipeline structures [32,33,34]. Zhao et al. [35] explored the pipeline crack monitoring theory and the piezoelectric ultrasonic time reversal method to locate the circumferential position of pipeline defects. Du et al. [36] examined the feasibility of a pipeline corrosion pit monitoring experiment based on time reversal. They utilized the peak amplitude of the focused signal to quantitatively evaluate pipeline corrosion status. However, the damage information of the structure obtained only by intuitively analyzing the waveform characteristics of the focused signal is very limited, and it is impossible to accurately classify and assess the corrosion degree of the pipeline.

The manual feature extraction work depends on high expertise and costs too much labor. Moreover, the final obtained features are not always effective when faced with unknown working conditions or application scenarios [37]. The Convolutional Neural Network (CNN) is one of the representative algorithms of deep learning, which can automatically extract waveform features and can learn inherent features of signals to perform more accurate classification and identification of fault diagnosis [10]. In recent years, CNNs have shown great potential in structural damage identification. Yang et al. proposed a vision-based automated method for the identification of the surface condition of concrete structures, which consisted of state-of-the-art pre-trained CNNs, transfer learning, and decision-level image fusion [38]. The proposed method was capable of accurately pinpointing the crack profile with wrong predictions of limited areas. Yang et al. improved the bird swarm algorithm to optimize the two-dimensional CNN, and the improved algorithm exhibited great performance in predicting the torsional strength of reinforced concrete (RC) beams [39]. Guo et al. developed a hierarchical CNN that extracted features automatically from raw vibration data and diagnosed bearing faults plus severity at the same time [40]. Peng et al. put forward a deeper residual one-dimensional CNN for adaptively learning fault features of the original vibration signal and obtained very high diagnostic accuracy for the fault diagnosis of wheelset bearings in high-speed trains [41]. However, if a CNN is used directly to classify original signals, the training of CNNs will become very challenging as the size of the original signal dataset increases. This will also give rise to an increase in training time and a higher requirement for model structure [42].

In this paper, a novel pipeline corrosion monitoring method based on wavelet packet energy (WPE) and CNN is proposed, which is denoted as WPE-CNN. Compared with other diagnostic methods, wavelet packet transform can continue to decompose the high-frequency components that are not decomposed by wavelet transform and extract the high-frequency features of the signal. Through this approach, the division of the signal in the full frequency band range is realized, and a more refined analysis of the signal is obtained [43]. The characteristic information of the original signal exists in each sub-band signal, and the signal characteristics can be further extracted by analyzing and determining the energy of each band. By using the sub-band energy obtained after wavelet packet decomposition as the input of the CNN model, the size of the data samples will also be diminished by a large proportion, thus reducing the complexity of the CNN model. Therefore, the WPE-CNN can not only further extract features and improve the accuracy of classification and recognition but also can greatly reduce the complexity of the CNN model and improve the efficiency of model training.

The piezoelectric active sensing with time reversal is utilized in this paper to quantitatively monitor the internal corrosion of pipelines. The specific arrangement of the experiment is as follows: First, the pipeline is corroded to different depths by electrochemical corrosion. After different corrosion depths on the PZT, the focused signal is collected by the time reversal method and is decomposed into sub-band energy by the wavelet packet. Then, a CNN is employed to convolve the wavelet packet energy, and the pipelines with different corrosion degrees are identified according to the output matrix. The rest of this paper is organized as follows: Section 2 explains the monitoring principle of the time reversal method and the classification principle of pipeline internal corrosion degree based on wavelet packet energy and the CNN model. Section 3 describes the experimental equipment, procedures, and results. Section 4 discusses the experimental results and compares them with other identification models. Section 5 summarizes and assesses the proposed method.

## 2. Materials and Methods

### 2.1. Time Reversal Technology

In applications, pipes usually work in a high-noise environment. Therefore, how to eliminate noise is the focus of pipeline corrosion monitoring. Time reversal can form a focus in a specific space and effectively improve the signal-to-noise ratio [44]. Therefore, the piezoelectric combined with the time reversal method can be used to monitor pipeline corrosion well. In this paper, two piezoelectric ceramic patches, labeled PZT1 and PZT2, installed on the surface of the pipeline, are used as actuators and sensors, as shown in Figure 1.

In the first step, the excitation signal generated by PZT1 is expressed as *x*(*t*) = *Bδ*(*t*). The corresponding pulse response function from PZT1 to PZT2 is expressed as *h* (*t*). Then the response signal of PZT2 is
(1)$$y_{1}\left(t\right)=x\left(t\right)\times h\left(t\right)=Bh\left(t\right)$$
where *B* is the coefficient of the amplitude of the excitation pulse signal. Then, in the second step, the response signal is reversed in the time domain:(2)$$y_{1}\left(-t\right)=Bh\left(-t\right)$$

Then, the reversed signal *y*1(−*t*) is applied to PZT1 as excitation again, and the focused signal *y*2(*t*) of PZT2 is
(3)$$y_{2}\left(t\right)=Bh\left(-t\right)\times h\left(t\right)=BR_{h}\left(t\right)$$
where *R_h_*(*t*) is the autocorrelation function of *h*(*t*).

According to the property of the autocorrelation function, when *t* = 0, the autocorrelation function *R_h_*(*t*) has the maximum value:(4)$$y_{2}\left(0\right)=B\int_{-\infty }^{\infty }h^{2}\left(\tau \right)d\tau =\frac{1}{B}\int_{-\infty }^{\infty }y_{1}^{2}\left(t\right)dt$$

According to the above derivation and analysis, the spectrum of the autocorrelation function is the square of the spectrum of the original function. That is, after time inversion, the difference between the signal spectrum will be amplified, and the signal features will be easier to extract.

### 2.2. WPE-CNN Operating Principle

The method combining WPE and CNN proposed for pipeline corrosion identification mainly includes signal energy feature extraction and CNN classification. The structure of the method is shown in Figure 2

First, the focused signals acquired by PZT2 after time reversal are decomposed into wavelet packets to obtain the energy characteristics of different frequency bands. Then, the data are divided into the training set, the validation set, and the testing set. The training data are input to CNN for training to obtain the classification model. Finally, the test data are input to the trained model for classification to obtain the final classification result.

### 2.3. WPE Operating Principle

Wavelet packet decomposition (WPD), a complete level-by-level decomposition of the original signal, aims to split the signal into successive low- and high-frequency components using a recursive filter-decimation operation, which has been proven to be a useful tool for non-stationary signal analysis [37]. The binary tree up to the third level wavelet packet decomposition of the original signal is shown in Figure 3.

After each WPD, the characteristic information of the original signal is also present in each sub-band signal. Therefore, the distribution characteristics of the energy distributed in different sub-bands can be used as an important basis for signal identification.

Suppose the length of the original signal is *N*, then the data length of the discrete signal in the decomposed signal is reduced to 2^−*k*^
*N*, and the energy of the sub-band signal after the wavelet packet decomposition can be expressed as
(5)$$E\left(x^{k,m}(i)\right)=\frac{1}{2^{-k}N-1}\sum_{i=1}^{2^{-k}N}{\left(x^{k,m}(i)\right)}^{2}$$
where *k* is the number of decomposition; *m* = 0, 1, 2, …, 2 *k* − 1 represents the position number of the decomposition frequency band.

The energy of the sub-band signals after wavelet packet decomposition is obtained and normalized.
(6)$$E_{m}=\frac{E\left(x{\left(t\right)}^{k,m}\right)-E{\left(x{\left(t\right)}^{k,m}\right)}_{min}}{E{\left(x{\left(t\right)}^{k,m}\right)}_{max}-E{\left(x{\left(t\right)}^{k,m}\right)}_{min}}$$*E*(*x*(*t*)*^k,m^*)*_max_* is the maximum value of the energy signal in the decomposed band. *E*(*x*(*t*)*^k,m^*)*_min_* is the minimum value of the energy signal in the decomposed band. *E_m_* is the energy value after normalization.

Remarkably, there are two crucial issues for achieving the best performance in the wavelet packet transformation: determining the optimal mother wavelet function, which is the first wavelet, and choosing the proper decomposition level of the signal. The value of the norm entropy *l^p^* calculated by the cost function in Equation (7) acts as the selection criteria to tackle the two issues mentioned above.
(7)$$l_{p}=\sum_{i}|E\left(x{\left(t\right)}^{k,m}\right){|}^{p}1\leq p\leq 2$$

### 2.4. CNN Operating Principle

The structure of CNN in this paper is composed of an input layer, two convolution layers, two pooling layers, a full connection layer, a dropout layer, and an output layer. Among them, the function of the convolution layer is to adaptively extract the energy of sub-band signals after wavelet packet decomposition [45]. Adding a pooling layer after the convolution layer can reduce the sampling of the input features, extract significant features, and reduce the parameters that need training while retaining the dominant features [46]. Then, the dropout layer is used to prevent overfitting. Finally, the features extracted from the convolution layer and pooling layer are synthesized by connecting the neurons of the full connected layer in pairs.

The super parameters of the Convolutional Neural Network directly affect the efficiency and accuracy of the model, which needs to be adjusted by experience and repeated experiments. After many tests, the Convolution Neural Network model in this paper is shown in Table 1. The prediction results are output through the softmax classifier. In the CNN model proposed in this paper, the iterations are 200, the learning rate is 0.001, the dropout rate is 0.1, the activation function is Relu, and the loss function is categorical cross-entropy.

The performance of the final trained CNN model needed to be evaluated by corresponding metrics [47]. Common evaluation metrics for classification tasks are true positive rate (*TPR*) and false positive rate (*FPR*) [48,49], which have the following equations:(8)$$TPR=\frac{TP}{TP+FN}$$
(9)$$FPR=\frac{FP}{FP+TN}$$
where *TP* indicates that a positive sample is correctly identified as a positive sample, *TN* indicates that a negative sample is correctly identified as a negative sample, *FP* indicates a false positive sample (which means that a negative sample is incorrectly identified as a positive sample), and *FN* indicates a false negative sample (which means that a positive sample is incorrectly identified as a negative sample).

## 3. Experimental Setup and Procedures

To verify the effectiveness of the proposed method, an experimental study was carried out. The experimental setup consists of a sample pipeline with two PZT patches mounted on the surface as actuator and sensor, respectively, a multifunctional data acquisition device (NIUSB-6361), and a laptop computer. The PZT patch was pasted on both sides of the outer surface of the pipeline sample using a quick-curing epoxy resin (brand name J-B WELD) to protect the pipeline from water corrosion.

To simulate the occurrence and evolution of pipeline internal corrosion, Figure 4 shows the electrochemical corrosion settings designed in this paper. The main parameters of experimental materials and devices are shown in Table 2. The graphite rod and sample pipeline were connected to the anode and cathode of an external DC power supply (China Delixi Electric). In addition, the output voltage of the electrochemical DC power supply was 12 V, and the output current was set to a constant value of 3 A. A 10%NaCl solution was used as the electrolytic solution.

The corrosion monitoring test was performed every 5 h after the pipeline corrosion. The total corrosion time was 40 h. Therefore, 9 corrosion states were obtained, including no corrosion (0 h). The corrosion states are shown in Table 3.

After the corrosion every 5 h, the pipeline was cleaned and the inner diameter of the pipe wall was measured with a gauge caliper 323-134-95 mm (Three quantities, China) with a precision of 0.01 mm. The inner diameter difference between adjacent corrosion periods is the corroded wall thickness per unit of time.

The Control and Data Acquisition (DAQ) program in LabVIEW was run on the computer to generate the pulse signal. The parameters of the pulse signal are shown in Table 4. The NIUSB-6361 converts the pulse signal into an analog signal to drive the PZT1. Subsequently, the signal propagates along the pipeline wall, and PZT2 acquires the response signal, which is reversed in the time domain. Then, the reversed signal is re-emitted, and the focused signal detected by PZT2 is acquired by the data acquisition (DAQ) program. The focused signals under different corrosion degrees are automatically stored by LabVIEW programming software.

## 4. Experimental Results

### 4.1. Data Acquisition

The corrosion process of the sample pipeline from 0 to 40 h is shown in Figure 5. The relationship between the pipeline corrosion time and the corrosion wall thickness is shown in Figure 6. With the increase in corrosion time, the corrosion depth gradually increases.

The focused signals with nine corrosion degrees received by PZT2 are shown in Figure 7. However, the nine focused signals with different degrees of corrosion are slightly different in the time domain. It is impossible to classify the corrosion degree of the pipeline intuitively.

### 4.2. Wavelet Packet Energy Feature Extraction

The foremost task in the wavelet packet analysis is to make a choice of the most appropriate mother wavelet function, since the results of signals from different functions could be varied [50]. The Daubechies family (dbN), exhibiting the properties of asymmetric, orthogonal, and biorthogonal, was adopted as the candidate mother wavelet function in this case. The norm entropy *l^p^* calculated from Equation (7) helps to determine the most appropriate order N of the Daubechies family. The smaller the value of the norm entropy *l^p^* is, the better the order is. Moreover, the smaller norm entropy *l^p^* will lead to less calculation time. Table 5 lists the norm entropy *l^p^* of the candidate Daubechies families dbN, where the order N is from 1 to 9. As shown clearly in Table 5, db5 results in the minimum of the cost function, which can be regarded as the best option for the mother wavelet function in this case [51,52].

Table 6 lists the value of the norm entropy *l^p^* after the wavelet packet decomposition at various decomposition levels from 1 to 9. As seen in Table 6, the db5 under five levels of decomposition obtains the smallest *l^p^* compared to the others. In other words, the modeling performance under five levels of decomposition is better than others. Consequently, the Daubechies with order N = 5 was applied as the mother wavelet packet, which was decomposed up to five levels in this case.

The energy of sub-band signals of 32 bands in the fifth layer was extracted from the low-frequency sub-band to high-frequency sub-band to obtain a set of 32-dimensional vectors. Through the feature extraction of wavelet packet energy, the differentiations of nine signals with different corrosion degrees were improved. However, the difference is still not obvious and difficult to distinguish intuitively, as shown in Figure 8.

The signal energy is mostly concentrated between 10–25 segments, and there is almost no energy distribution between 0–10 and 25–32 signals. Since wavelet packet components with small energy are easily interfered by signal noise, these components can be ignored. It is also reasonable to take only 10–25 segments of wavelet packet components with large energy to approximate the original signal. Therefore, this paper intercepted 16 segments of the signal sub-band energy between 10 and 25 as the input vector of the CNN model, which can further improve the training efficiency and classification accuracy.

### 4.3. Evaluation of WPE-CNN Model for Different Corrosion Degrees

The 16 segments of sub-band energy after wavelet packet decomposition were used as input vectors in the WPE-CNN model proposed in this paper for classifying nine kinds of pipeline corrosion degrees. There are 100 datasets for each corrosion degree, 60% of which were randomly selected as the training set, 20% as the verification set, and 20% as the testing set.

The training process of the WPE-CNN model for the pipeline corrosion degree is shown in Figure 9. After about 46 iterations, accuracy and loss values of the training and validation datasets were very stable, and the model started to converge. The accuracy rate reached 99.4% and the value of the loss function finally stabilized at around 0.056.

Under different corrosion degrees, Figure 10 shows the prediction results of the WPE-CNN model. The accuracy of each sample is 100%, 100%, 100%, 100%, 94.7%, 100%, 100%, 100%, and 90.9%. The total accuracy of the nine states is 98.30%. The method proposed in this paper has high accuracy in classifying the internal corrosion degree of pipelines.

### 4.4. Model Comparison

For comparison, the data after feature extraction were input to other commonly used models for classification, such as the SVM model and KNN model. The SVM process was performed with Python, with RBF as the kernel function; the parameter coefficient g of the kernel function was 0.625 and the penalty factor coefficient c was 0.6. K was 12 in the KNN process. In addition, the original data without feature extraction were input into the CNN model for classification.

To reduce experimental errors and improve the accuracy of experimental conclusions, the WPE-CNN, CNN, the WPE-SVM, and the WPE-KNN were trained 20 times to verify the accuracy of test samples, as shown in Figure 11. The final results show that the recognition accuracy of the WPE-CNN is between 98.12% and 99.40%, and the training time is 4.78 s. The recognition accuracy of the CNN is between 90.33% and 93.38%. The recognition accuracy of the WPE-SVM is between 88.13% and 91.30%. The recognition accuracy of the WPE-KNN is between 87.43% and 90.05%. The standard deviation of the accuracy of the WPE-CNN is 0.4533 in 20 repetitional trainings, which is the smallest between four methods. This fully indicates that the method proposed in this paper can effectively improve the identification accuracy of pipeline corrosion degree and has better stability than the CNN, WPE-SVM, and WPE-KNN.

The training times of the four models are shown in the Table 7. Compared with the CNN, the training efficiency of the WPE-CNN model was improved hundreds of times. Compared with the WPE-SVM and WPE-KNN, the WPE-CNN effectively improved the classification accuracy while consuming limited training time.

## 5. Conclusions

In this paper, a new method based on time reversal combining WPE and CNN is proposed to quantitatively identify pipeline internal corrosion. The proposed method has low cost, simple structure, short time consumption, and high accuracy. The major findings of the proposed approach can be summarized as follows:
The way of quantitatively monitoring the internal corrosion of pipelines by piezoelectric active sensing with time reversal can be considered as a novel and cost-effective approach.The actual output of the WPE-CNN is basically consistent with the theoretical output during the proposed approach. The proposed method can effectively shorten the training time and greatly improve the recognition accuracy.The recognition average accuracy of the WPE-CNN reaches 99.01%, which can be effectively used for pipeline internal corrosion monitoring.The recognition accuracy of the WPE-CNN is 98.12~99.40%, and the training time of the WPE-CNN is 4.78 s. Compared with other methods based on classical deep learning models, this method has higher diagnosis accuracy, faster training speed, and more stable performance.

The experimental results show that it is feasible and effective to use the WPE-CNN recognition model, based on the time reversal method, to classify the internal corrosion degree of the pipeline. To further develop this new method, different lengths and diameters of pipelines will be selected for comparative experiments. The types of corrosion of pipelines, such as pitting and regional corrosion, will also be addressed in the following studies.

## Figures and Tables

**Figure 1 sensors-23-00855-f001:**
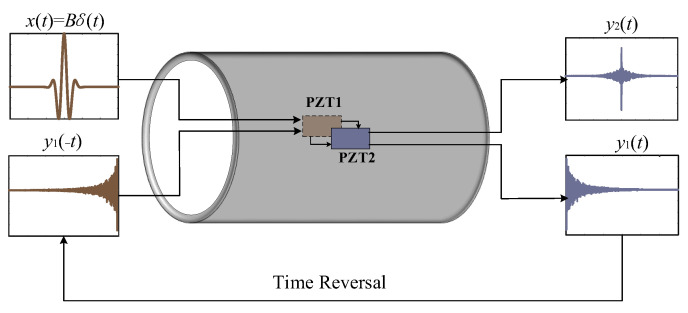
Time reversal process.

**Figure 2 sensors-23-00855-f002:**
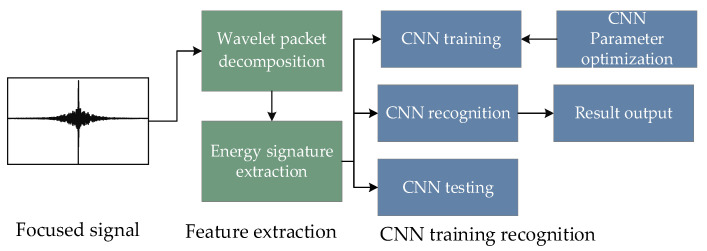
Structure of WPE-CNN.

**Figure 3 sensors-23-00855-f003:**
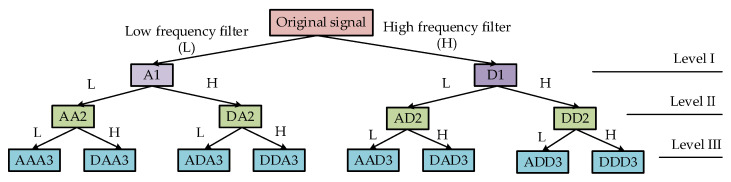
Structure of wavelet packet transform.

**Figure 4 sensors-23-00855-f004:**
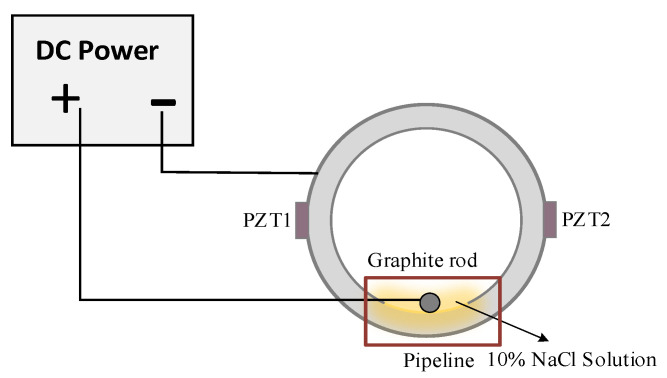
Electrochemical corrosion device.

**Figure 5 sensors-23-00855-f005:**
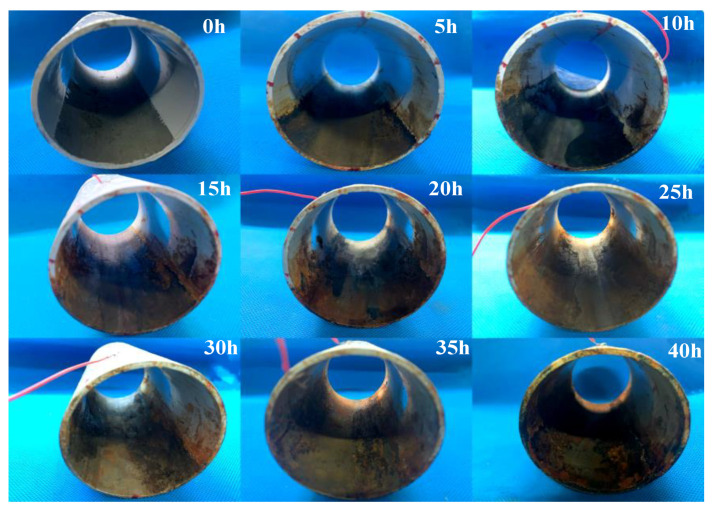
Pipeline status under different operating conditions.

**Figure 6 sensors-23-00855-f006:**
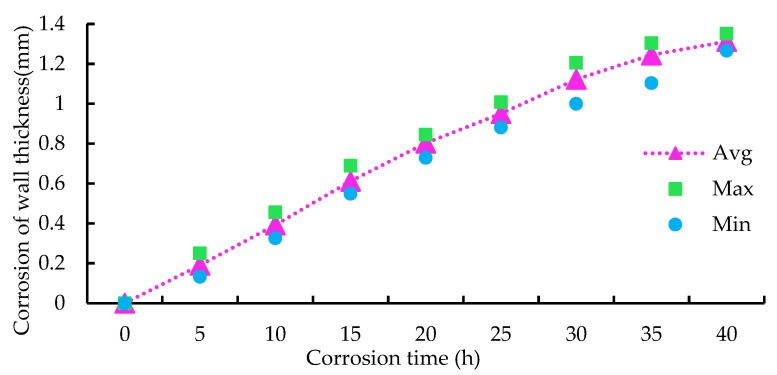
Relationship between the wall thickness and the corrosion time.

**Figure 7 sensors-23-00855-f007:**
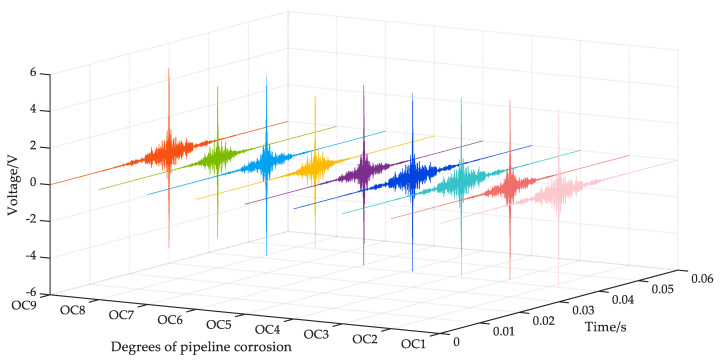
Focused signals under nine operating conditions.

**Figure 8 sensors-23-00855-f008:**
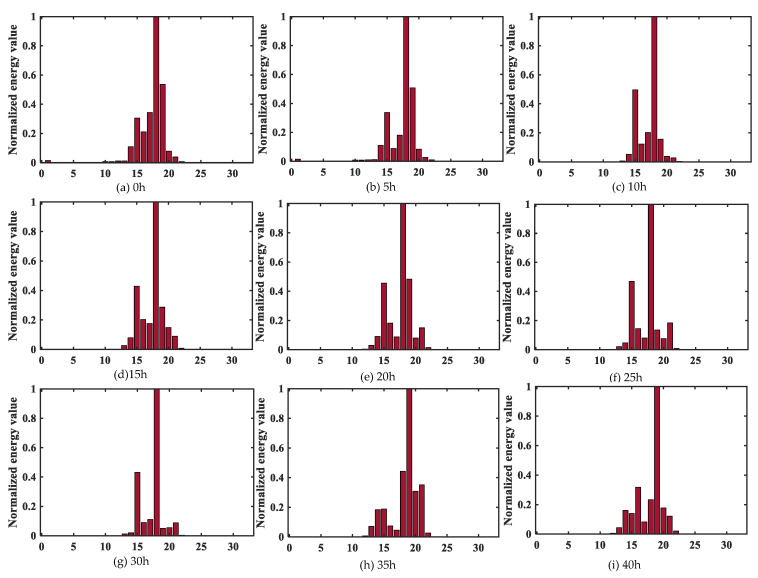
Wavelet packet energy characteristics under different corrosion degrees.

**Figure 9 sensors-23-00855-f009:**
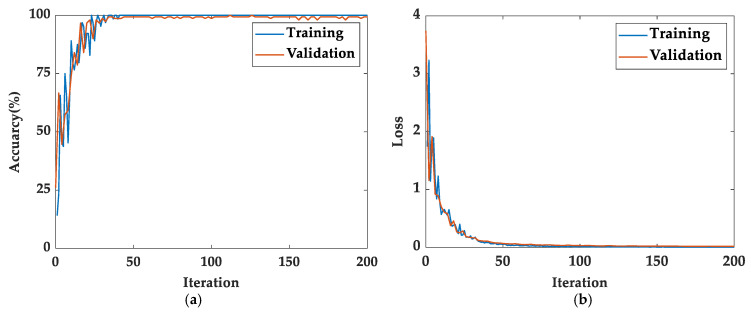
The WPE-CNN model training process: (**a**) accuracy (%); (**b**) loss.

**Figure 10 sensors-23-00855-f010:**
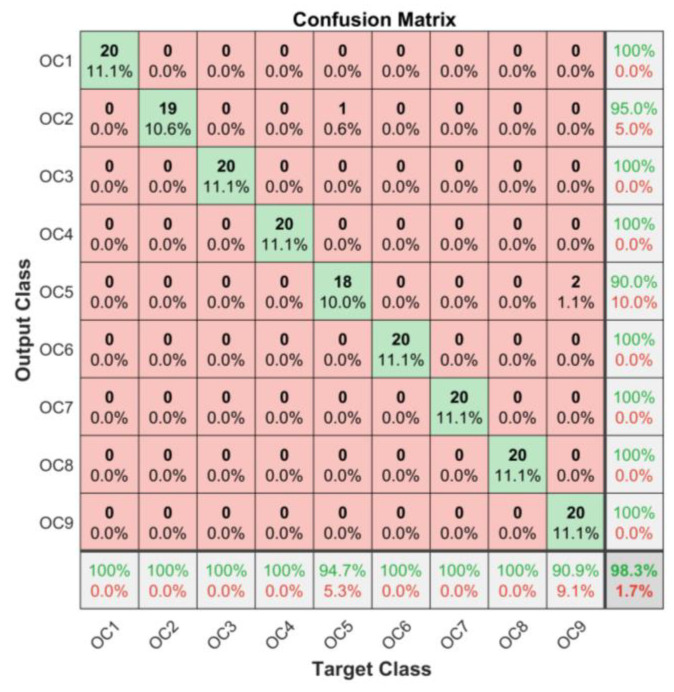
Confusion matrix of test results.

**Figure 11 sensors-23-00855-f011:**
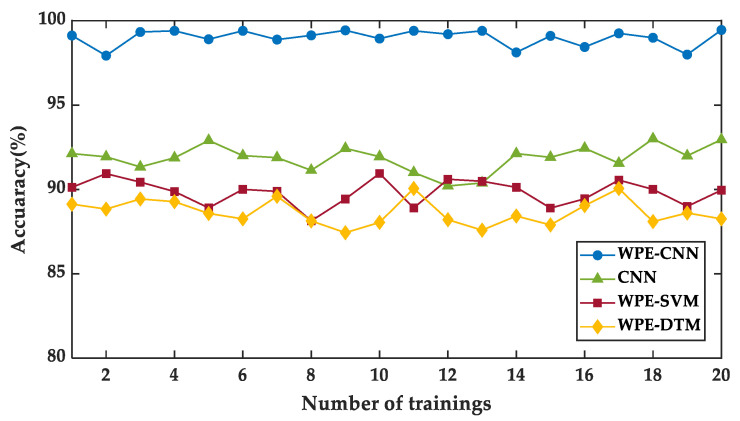
Comparison of classification accuracy with three models.

**Table 1 sensors-23-00855-t001:** WPE-CNN structure parameter setting.

Network Layer	Data Length @ Number of Channels	Kernel Length @ Number of Kernels	Length of Pooling
Input	16 @ 1		
Convolution layer	16 @ 32	3 @ 32	
Pooling layer	8 @ 32		2
Convolution layer	8 @ 32	3 @ 32	
Pooling layer	4 @ 32		2
Fully connected layer	128		
Softmax	9		

**Table 2 sensors-23-00855-t002:** Sample pipeline and experimental device-related parameters.

Components	Parameters	Values	Unit
PZT transducers	Density	7500	Kg/m^3^
	Young’s modulus	6.67 × 10^10^	N/m^2^
	Piezoelectric strain coefficients	−280/620/860	10^−12^ m/V
	Dielectric constants	3200	-
	Dimensions	10 × 4 × 1 (±0.1)	mm^3^
Q235 Steel pipeline	Outer diameter	88	mm
	Inner diameter	82	mm
	Length	200	mm
	Density	7850	kg/m^3^
	Elastic modulus	205,000	Mpa
	Poisson ratio	0.30	-
Graphite rod	Diameter	6	mm
	Length	100	mm
NaCl Solution	Volume	400	mL

**Table 3 sensors-23-00855-t003:** Testing operating conditions.

Operating Conditions	OC1	OC2	OC3	OC4	OC5	OC6	OC7	OC8	OC9
Corrosion time (h)	0 h	5 h	10 h	15 h	20 h	25 h	30 h	35 h	40 h

**Table 4 sensors-23-00855-t004:** Parameters of the pulse waveform.

Parameter	Value	Unit
Amplitude	10	V
Center frequency	150	kHz
Sample frequency	1	MHz
Normalized bandwidth	0.8	-

**Table 5 sensors-23-00855-t005:** Norm entropy *l^p^* in different order N of the Daubechies wavelet family.

Order N	1	2	3	4	5	6	7	8	9
Norm entropy *l^p^*	395.33	377.38	362.70	354.61	334.61	382.80	400.80	395.61	406.59

**Table 6 sensors-23-00855-t006:** Norm entropy *l^p^* in different decomposition with db5 wavelet.

Level j	1	2	3	4	5	6	7	8	9
Norm entropy *l^p^*	580.54	497.03	426.16	365.51	313.99	380.38	367.33	352.6352	431.26

**Table 7 sensors-23-00855-t007:** Comparison of training time and accuracy of three models.

Method Name	Training Time (s)	Accuracy (%)			
		Max	Min	Mean	Std
WPE-CNN	4.78	99.40	98.12	99.01	0.4533
CNN	415.14	93.38	90.33	92.10	0.7612
WPE-SVM	2.64	91.30	88.13	89.85	0.7644
WPE-KNN	2.36	90.05	87.43	88.64	0.7592

## Data Availability

Due to the nature of this research, participants of this study did not agree for their data to be shared publicly, and therefore these data are only available upon reasonable request.

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
