# Peer review of "A Novel Pipeline Corrosion Monitoring Method Based on Piezoelectric Active Sensing and CNN"

_sensors, 2023, doi:10.3390/s23020855_

Round 1

Reviewer 1 Report

Comments to the authors:

1. kindly add the %classification accuracy in the abstract.

2. In the introduction section, kindly avoid the group citation.

3. The recent literature in the field of pipeline corrosion monitoring has to be updated in the article.

4. The research gap and novelty are not clear.

5. The authors selected CNN as the classification algorithm. The results have to be compared with other classification algorithms like DT, KNN, MLP etc.

6. The same methodology has to be tested with any standard benchmarking datasets for validation.

7. The authors have to ensure that any overfitting?. CNN yields 100% accuracy is not possible.

8. All the equations have to be cited properly.

9. Section 2.4 requires literature support.

10. In section 3, kindly mention sensor specifications (Sensitivity) and sample rate.

11. Figure 7 can be represented in a better way. in this current form unable to visualize the signals for all 8 conditions and data for 10h is missing.

12. Normally 70:30 or 80:20 is used for training and testing. The authors selected 50:50. Is Any specific reason, that may lead to overfitting?

13. The TPR and FPR values have to be addressed along with the confusion matrix.

14. A separate discussion section is needed in the article to bring the essence of the research.

15. In Figure 11, within 20 times of training, the results are not converged.

16. Compared to other algorithms, CNN consumes more time. Why?

17. Language and style have to be improved.

Reviewer 2 Report

This manuscript proposed a novel piezo-electric active sensing-based approach for pipeline corrosion monitoring using deep learning technology, where convolutional neural network (CNN) was utilised for the task of interest. First of all, the wavelet packet transform was employed to extract signal features. Then, CNN model was developed for corrosion degree identification. Finally, the performance of the proposed method was validated using experimental data, with satisfactory results. Overall, the topic of this research is interesting, and the manuscript was well organised and written. The detailed comments are presented below.

1.       The main findings and innovation of this paper should be highlighted in both abstract and introduction.

2.       Broaden and update literature review on deep learning or CNN for structural condition/defect assessment. E.g. Torsional capacity evaluation of RC beams using an improved bird swarm algorithm optimised 2D convolutional neural network; Vision-based concrete crack detection using a hybrid framework considering noise effect.

3.       Please give more explanations on why db5 was selected as mother wavelet and decomposition level was 5.

4.       The architecture of the proposed CNN should be clearly presented. Also network hyperparameter setting

5.       The author only considers the training and validation for CNN development (results shown in Fig.9). How about the testing data for model evaluation?

6.       Fig. 11: The presented CNN was compared with SVM for model performance evaluation. The details on SVM model should be provided.

Reviewer 3 Report

Please see the reviewer's comments in the pdf attached file. Thanks.

Round 2

Reviewer 1 Report

All the best to the authors.

Reviewer 3 Report

Please fix the not-found references before publication.